# Low Represented Mutation Clustering in SARS-CoV-2 B.1.1.7 Sublineage Group with Synonymous Mutations in the E Gene

**DOI:** 10.3390/diagnostics11122286

**Published:** 2021-12-07

**Authors:** Paolo Giuseppe Bonacci, Dalida Angela Bivona, Dafne Bongiorno, Stefano Stracquadanio, Mariacristina Massimino, Carmelo Bonomo, Alessia Stracuzzi, Paolo Pennisi, Nicolò Musso, Stefania Stefani

**Affiliations:** 1Department of Biomedical and Biotechnological Science (BIOMETEC), University of Catania, Via Santa Sofia 97, 95123 Catania, Italy; paolo.g.bonacci@gmail.com (P.G.B.); dalidabivona@gmail.com (D.A.B.); s.stracquadanio@unict.it (S.S.); m.cri1503@gmail.com (M.M.); carmelobonomo94@gmail.com (C.B.); stefanis@unict.it (S.S.); 2LifeGene Società a Responsabilità Limitata, Via Garibaldi 377, 98121 Messina, Italy; ale.stracuzzi@hotmail.it (A.S.); penpaolo11@gmail.com (P.P.); 3Centro Diagnostico Ionia (CDI), Via Cavour 11, Riposto, 95018 Catania, Italy

**Keywords:** E gene, molecular testing, Next Generation Sequencing

## Abstract

Starting in 2019, the COVID-19 pandemic is a global threat that is difficult to monitor. SARS-CoV-2 is known to undergo frequent mutations, including SNPs and deletions, which seem to be transmitted together, forming clusters that define specific lineages. Reverse-Transcription quantitative PCR (RT-qPCR) has been used for SARS-CoV-2 diagnosis and is still considered the gold standard method. Our Eukaryotic Host Pathogens Interaction (EHPI) laboratory received six SARS-CoV-2-positive samples from a Sicilian private analysis laboratory, four of which showed a dropout of the E gene. Our sequencing data revealed the presence of a synonymous mutation (c.26415 C > T, TAC > TAT) in the E gene of all four samples showing the dropout in RT-qPCR. Interestingly, these samples also harbored three other mutations (S137L—Orf1ab; N439K—S gene; A156S—N gene), which had a very low diffusion rate worldwide. This combination suggested that these mutations may be linked to each other and more common in a specific area than in the rest of the world. Thus, we decided to analyze the 103 sequences in our internal database in order to confirm or disprove our “mutation cluster hypothesis”. Within our database, one sample showed the synonymous mutation (c.26415 C > T, TAC > TAT) in the E gene. This work underlines the importance of territorial epidemiological surveillance by means of NGS and the sequencing of samples with clinical and or technical particularities, e.g., post-vaccine infections or RT-qPCR amplification failures, to allow for the early identification of these SNPs. This approach may be an effective method to detect new mutational clusters and thus to predict new emerging SARS-CoV-2 lineages before they spread globally.

## 1. Introduction

The COVID-19 pandemic presents a challenge to the resilience of global healthcare [1]. To defuse the threat posed by the severe respiratory syndrome coronavirus-2 (SARS-CoV-2), responsible for the pandemic, health agencies from all over the world (including the World Health Organization, WHO) suggest sequencing as many viral samples as possible [2,3,4]. This is the only way to track the virus spread and epidemiology as well as to try to prevent recurrent pandemic waves.

In a recent study, Rausch et al. showed that SARS-CoV-2 has a lower mutation rate than other human viral pathogens, which leads it to incorporate only mutations beneficial for its fitness [5]. Some mutations seem to be transmitted together, forming clusters that define specific lineages or “variants” [6,7]. Sequencing and computing consensus sequences or genomes from RNA viruses such as the SARS-CoV-2 present challenges due to their evolution rate, which is considerably higher than that of their hosts [8]. To date, the most common sequencing method employed with SARS-CoV-2 is a combination of PCR-based amplification and Next Generation Sequencing (NGS) [9]. Bioinformatic tools and databases are very helpful for researchers, although—unfortunately—not all countries have a sufficiently significant number of sequencing laboratories, and not all sequenced genomes are uploaded to databases [10]. The most employed tools used to report all useful information about the different viral clades and suitable to analyze and classify the variant of the sequenced SARS-CoV-2 strains are Phylogenetic Assignment of Named Global Outbreak Lineages (PANGOLIN, cov-lineages) [11] and Nextstrain [12]. Pangolin allows a user to assign a SARS-CoV-2 genome sequence the most likely lineage to SARS-CoV-2 query sequences, whereas Nextstrain CoVariants provides an overview of SARS-CoV-2 variants and mutations that are of interest, specifying what mutations define a variant, what impact they might have, and where variants are found. To date, most of the resulting sequences are submitted to the GISAID database [13,14]. The analyses of SARS-CoV-2 genome sequences led to clustering isolates based on mutations and informed researchers on how rare a cluster of mutations is—all around the world as well as in specific regions. However, the significant mutation rate observed for SARS-CoV-2 makes it very hard to define what a cluster is. It is well known that one of the main mutational hotspots is the S gene encoding for the Spike protein, involved, along with host surface receptors, in the entry of the virus into the host cell, [15] but when analyzing the sequences included in databases, it seems clear that, in some cases, even well-known widespread variants (e.g., Alpha, Beta, Gamma, and Delta) have certain sets of peculiar mutations as well as random, often synonymous, mutations. Another important conserved structural gene that was recently shown to harbor “extra-lineage mutations “is the E gene. The SARS-CoV-2 E protein is mainly involved in viral envelope formation, but also in pathogenic mechanisms. In particular, the C-terminal domain of the monomeric E protein was shown to interfere with intracellular compartments such as the endoplasmic reticulum and the Golgi [16].

In this scenario, our Eukaryotic Host Pathogens Interaction (EHPI) research Group received six SARS-CoV-2 samples collected by an analysis laboratory in the area of Eastern Sicily between June and July 2021. Four out of six samples showed a dropout of the E gene—i.e., a failure in its amplification [17,18,19]—likely due to the presence of a mutation in the E gene probe annealing sequence, the more so as their RNAs had already been tested by RT-qPCR using commercial kits for lineage identification. Thus, the samples were sequenced, which revealed the presence of a synonymous mutation (c.26415 C > T, TAC > TAT, Reference Genome: NC_045512) in the E gene and three other mutations (S137L—Orf1ab gene, N439K—Spike gene, and A156S—Nucleocapsid gene) scattered throughout the genome in all samples carrying the E gene dropout. Notably, these three mutations had a very low diffusion rate worldwide but were found in all our samples together with the synonymous E mutations. This particular combination suggested that these mutations may be somewhat linked to each other and more common in a specific area than in the rest of the world, leading us to look back to all the 103 sequences we had already analyzed in our internal database to see whether our hypothesis of a “cluster” of otherwise rare mutations could be confirmed.

The relevance of this hypothesis is to be considered in the context of epidemiological surveillance and its evolution since the beginning of the pandemic, moving from a substantially “subjective” approach, highly influenced by local specificities and therefore highly variable, to a more organized one with increasing understanding of the virus. Against this backdrop, genomic surveillance is still lagging behind. The reason for this is to be found in the parameters applied for the monitoring of the epidemiological curve, initially incidence and hospitalization rates, and now mainly cases of vaccinated persons with moderate to severe disease, which are known to be rather sporadic.

The goal of our work is to keep epidemiological surveillance active, especially at a local level. This is to be understood as an integrated microbiological and epidemiological surveillance that continuously and systematically compares and analyzes the information on a representative portion of SARS-CoV-2 cases at a regional level. The focus is, first of all, to support the health authorities in their decision-making, as well as to provide a useful and up-to-date tool that can be used for phylogenetic observations.

In this scenario, the collaboration between private laboratories, research centers, and hospitals plays a key role in achieving a comprehensive understanding of the SARS-CoV-2 genomic behavior through the comparison of the results obtained with different molecular approaches.

## 2. Materials and Methods

### 2.1. Samples Included in the Study

Six viral RNA samples that resulted as positive for SARS-CoV-2 were delivered to our laboratory by the “CDI Centro Diagnostico Ionia” in Riposto (Catania, Italy). Four samples caught out attention, as they showed an E gene dropout in RT-qPCR, while two were used as positive controls as they showed amplification of the E gene in RT-qPCR. (Figure 1). The analyses were performed using the MOLgen SARS-CoV-2 Real Time RT-PCR Kit (Adaltis S.r.l., 20122 Milan, Italy) following the manufacturer’s procedure.

Furthermore, the sequences of the six samples were analyzed using the following workflow: RNA Reverse Transcription; Designed Primers for E-Gene Amplification; Purification of PCR Products and Sanger Sequencing; Next Generation Sequencing (NGS) using the MiSeq Illumina^®^ Platform (Illumina, Inc., 92122, San Diego, CA, USA), then a comparison of the six sequenced genomes with those from our database.

### 2.2. Internal Database

The samples were delivered as nasopharyngeal swabs in inactivating solution or as already extracted RNA from public and private structures of Eastern Sicily. Given the large number of samples delivered to the EHPI laboratory, only some of them were selected for sequencing. Particularly, we decided to sequence samples from severely ill patients from different geographical areas or with altered amplification curves in RT-qPCR. Our internal database consisted of 103 viral samples sequenced by Next Generating Sequencing (NGS) as described below. Some parameters about the quality of the NGS run are reported in Table 1.

### 2.3. Primer Design, RNA Reverse Transcription and PCR Amplification

The primers used for the amplification of the E gene and a fragment of the S gene were designed on “MT215195 Severe acute respiratory syndrome coronavirus 2 ubicate SARS-CoV-2/human/HKG/90_VM20002907/2020 (complete genome sequence release date: 29 October 2020)” using the online software Primer3 Plus [20]. The reference sequence used to align the MT215195 sequences was downloaded using the BLAST tool [21].

RNA reverse transcription was carried out using the QuantiTect^®^ Reverse Transcription Kit (Ref. 205311, QIAGEN, 40724 Hilden, Germany), following the manufacturer’s instructions. To amplify the target region of the Spike gene where we recognized the SNPs useful to discriminate between lineages, we performed a PCR using the FS-4 forward and FS-5 reverse primers (Table 1), obtaining an amplicon of 1312 bp. To amplify the target region of the E gene, a primer pair was designed (Table 2).

PCRs targeting E and S genes of SARS-CoV-2 were performed using IllustraTM PuReTaqTM Ready-To-GoTM PCR Beads (Ref. 27955901, GE Healthcare, Chicago, IL, USA) to ensure the lowest possible levels of contaminating nucleic acids, 5 µL of cDNA; 0.5% of Dimethyl Sulfoxide (DMSO) (Ref. D8418, Sigma-Aldrich, Merck, Darmstadt, Germany), and 0.5 µM of each primer. The amplifications were performed with Eppendorf Mastercycler Nexus X2 with annealing at 55.5 °C for 1 min and extension at 72 °C for 1 min for 40 cycles. The obtained amplicons were verified as previously published [22].

### 2.4. Purification of PCR Products, SANGER Sequencing and Sequence Analysis

The amplicons obtained for the S and the E gene were purified with the ExoSAP-IT^®^ buffer (Ref. 78201, Thermo Fisher Scientific, Cleveland, OH, USA) according to the manufacturer’s protocol with some modification: the enzyme amount was increased by adding 3.5 μL of ExoSAP in 6.25 μL of PCR products. Purification, Sanger sequencing, and analysis were performed as previously described [23,24].

### 2.5. Next Generation Sequencing (NGS) on the MiSeq Illumina^®^ Platform

All the samples included in the study, the 103 viral RNAs that make up our internal database, and the six samples received from the CDI laboratory were sequenced with NGS technologies.

Library preparation for sequencing on the Illumina platform was performed using QIAseq DIRECT SARS-CoV-2 Kits (Ref. 333891, QIAGEN, Hilden, Germany) and the QIAseq^®^ FX DNA Library Core Kit (Ref. 1120146, QIAGEN, Hilden, Germany) following the manufacturer’s instructions.

Bead-based library purifications were performed using Agencourt^®^ AMPure^®^ XP Beads (Ref. A63880, Beckman Coulter^®^, Indianapolis, IN, USA). The libraries were quantified and evaluated for quality using the Qubit dsDNA HS Assay Kit (Ref. Q32851, Invitrogen, Carlsbad, NM, USA) and the Agilent^®^ High Sensitivity DNA Kit (Ref. 5067–4626, Agilent, Santa Clara, CA, USA). The libraries were denatured and diluted following the “MiSeq System Denature and Dilute Libraries Guide” (Illumina, Document #15039740 v10). The libraries were multiplexed with different barcodes and pooled at 4 nM in equimolar amounts. The pooled libraries were sequenced on Illumina MiSeq at a final concentration of 7.5 pM, using the MiSeq^®^ v2 reagent kit (Ref. 15033624, Illumina, San Diego, CA, USA). The mutational profiles of the six samples from the CDI were compared with those of the 109 samples in our internal database.

### 2.6. Bioinformatic Analysis

Data were analyzed and aligned using the QIAGEN CLC Genomics Workbench 21 software and following the User Manual for CLC Genomics Workbench 21.0.3, released on 25 January 2021 (QIAGEN Aarhus, Silkeborgvej 2 Prismet DK-8000 Aarhus C, Denmark). The evaluation of the impact of mutations at the post-translational protein level, the statistical frequency of the mutations, and the number of sequences globally available were obtained from the GISAID database [13]. In particular, the impact of mutations and their statistical frequency were evaluated by loading the FASTA files obtained from sequencing on the “CoVsurver: Mutation Analysis of hCoV-19” tool, made available by GISAID. The number of sequences from Italy and from all over the world, useful for the calculation of statistical frequency, was extracted from GISAID, while the sequences of the internal database were already available as these are used for ordinary sequencing experiments. Lineage assignment was performed with PANGOLIN [11].

## 3. Results

### 3.1. E Gene Mutation Analysis

Of the six samples from the CDI, four did not amplify for the E gene in RT-qPCR (Figure 1), whereas two were used as positive controls as they did amplify for the E gene. In any case, all samples tested positive for SARS-CoV-2. The outcomes of the sequencing in our laboratory reveal that all the samples that did not amplify in RT-qPCR harbor a synonymous mutation at the E gene level, namely in position c.2641 C > T (Reference Genome: NC_045512, TAC > TAT, Tyr > Tyr, Figure 2).

The outcome of sequencing allowed us to state that the original primer pair does not produce amplification in RT-qPCR if the aforementioned synonymous mutation (TAC > TAT) was present, as in the four remaining samples.

Once this result was obtained, we evaluated whether other samples with the same mutation (TAC > TAT) were present in our internal database.

Indeed, one sample from our database showed the presence of the synonymous mutation (TAC > TAT) at the E gene level. This sample had arrived at our laboratory with a request for urgent sequencing because the patient was returning from a trip to Africa.

The five samples with the synonymous mutation (four samples delivered by the CDI and one from our database) belonged to lineage B.1.1.7 (WHO label Alpha), but it was observed that they all shared some “extra-lineage” mutations: S137L in the Orf1ab gene, N439K in the Spike gene and A156S in the N gene (except for the African sample). The database made available by GISAID helped appraise the low frequency with which these mutations have been found worldwide (Table 3) and the important biological implications that the mutations themselves have for protein translation.

### 3.2. NGS Sequencing Analysis

After molecular characterization, we added the 6 CDI samples to our collection, obtaining an internal database of 109 viral RNAs from nasopharyngeal swabs sequenced via the MiSeq platform. The data obtained from the sequencing were analyzed using PANGOLIN, and the following lineages were detected (Figure 3): 83 samples belonging to lineage B.1.1.7 (Alpha); 9 samples belonging to lineage P.1 (Gamma); 12 samples belonging to lineage B.1.617.1 (Delta); 4 samples belonging to lineage B.1.525 (Eta); 1 sample belonging to lineage P.2 (Zeta, no longer considered as a variant of interest by the WHO [25]) (Figure 3).

Of the 75 samples belonging to lineage B.1.1.7, 20 showed at least one out of these three mutations: S137L (Open Reading Frame 1ab gene), N439K (Spike gene), and A156S (Nucleocapsid Gene). Of these, 5 harbored the synonymous mutation in the E gene at position 26415 bp, TAC > TAT, while the remaining 15 did not.

When focusing our attention on the mutational profiles of the 5 E-mutated patients, 4 of these carried all three above-described mutations, while one (the patient returning from Africa) did not have the A156S mutation.

As for the remaining 15 E-wild type patients, 14 carried the A156S mutation, and four out of these 14 patients also carried the S137L and N439K mutations, while the remaining 12 did not.

The remaining sample, negative for the synonymous mutations, was also negative for A156S and S137L, whereas it was positive for N439K only (Figure 3).

### 3.3. Statistical Analysis of Mutation Frequency

The frequencies of the four mutations were calculated in three different populations: the sequences available in our internal database, the sequences available in Italy, and, for comparison, all the sequences available worldwide. A sub-analysis of frequency was also conducted considering the samples belonging to lineage B.1.1.7 available in our internal database first, and then the mutated E gene samples only. All results in terms of statistical frequency are shown in Table 2. As a further step to support the statistical analysis, p-values were calculated by the χ2-test, again using the same three populations: our internal database (referred to as the EHPI Database), Italy, and World.

Correlation p-values were calculated considering the number of samples with or without a certain mutation (S137L, N439K, and A156S), each time comparing two populations, for a total of 3 combinations for each mutation: EHPI database vs. Italian database, EHPI vs. World database and Italy vs. World. The same calculation was then applied only to the 20 samples of our internal database that showed one of the three mutations of interest, comparing those that had the mutated E gene and those that had not. The results for correlation p-values are reported in Table 4.

## 4. Discussion

Although the SARS-CoV-2 pandemic has generated considerable global efforts in sequencing, there is still a mismatch between the number of SARS-CoV-2 genomes available per 100 cases, as shown by Colson and colleagues [26]. They showed that this observation is true not only for developing countries but also for developed ones. Often, the main reason for this gap is the lack of Next-Generation Sequencing instruments and/or the limited number of qualified human resources capable of using this advanced molecular technique.

As suggested by health agencies from all over the world, to defuse the threat posed by the severe respiratory syndrome coronavirus-2 (SARS-CoV-2), responsible for the COVID-19 pandemic, it is necessary to sequence as many viral samples as possible [2,3]. This is the only way to track the virus spread and epidemiology as well as to try to prevent recurrent pandemic waves.

Among sequencing methods, the NGS is the most important one because it is able to identify a genetic cluster of mutations before it generates a new lineage [27]. The massive and parallel sequencing system applied to many samples with particular characteristics allows both epidemiologic surveillance and the identification of a “new genetic hot spot SNPs”. SARS-CoV2 has a high mutation rate [28] for both synonymous and non-synonymous mutations. Many of these mutations are transient or have no role, and after a certain period of time, they can be found in a large part of the population. Some have a greater impact on biological functions, and, over time, they may add to other mutations resulting from the selective pressure [29] exerted by the virus circulating in a large number of hosts, creating clusters of driver mutations that may give rise to new lineages. In this sense, local epidemiological surveillance by means of NGS and the sequencing of samples with clinical and or technical particularities, e.g., post-vaccine infections or RT-qPCR amplification failures, allows for early identification of these SNPs. A clear link between a mutation in the E gene and three SNPs (S137L, N439K, and A156S) emerged from surveillance in Eastern Sicily. These mutational clusters probably originate from Southern Europe, this being supported by the presence of the aforementioned mutations in a sample collected from a patient returning from Africa. In this sense, unfortunately, there are not many NGS data coming from this area [30,31,32], which is why the cluster composed of S137L, N439K, and A156S is little expressed when comparing data against Northern Europe. If the significance of synonymous mutations within the mutation rate necessary for SARS-CoV-2 evolution is not yet clear, the functional significance of non-synonymous mutations is known in many cases; in two of the three mutations, the related E gene cluster plays a role in the antigenic drift as well as in surface receptor binding. Therefore, it is important to follow these mutations also through real-time “E dropout” in order to track their spread in the population. Frequently testing the entire world population for SARS-CoV-2 would be the ideal surveillance strategy for obtaining an actual insight on all asymptomatic and symptomatic individuals. Unfortunately, this is not a realistic strategy [33]. In this scenario, our work underlines the need to pay attention to samples showing technical particularities linked to mutations or clusters of mutations that escape identification with some of the available commercial kits, and which could be potentially involved in the emergence of new SARS-CoV-2 lineages. Identifying these kinds of mutations at an early stage, before an emerging lineage spreads worldwide, is a crucial point in genomic epidemiological surveillance. Moreover, the proposed approach could be applied on a regional basis, as extensive evidence has revealed that recurrent SARS-CoV-2 variants show variable mutation patterns within distinct geographic regions [34].

## Figures and Tables

**Figure 1 diagnostics-11-02286-f001:**
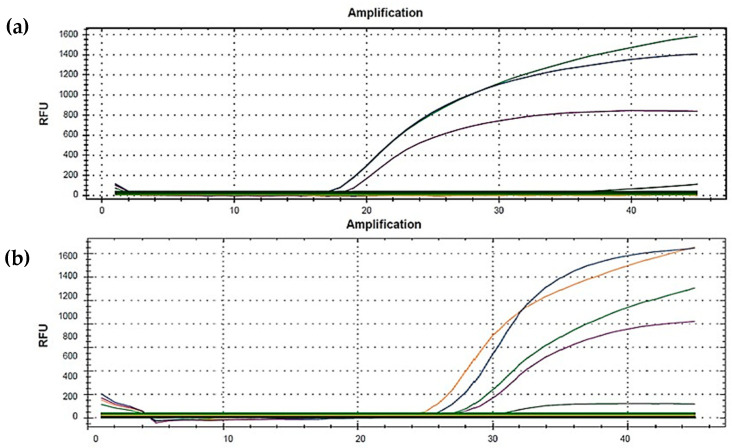
Amplification curves obtained by RT-PCR with the MOLgen SARS-CoV-2 Real Time RT-PCR Kit: (**a**) amplification curves of a sample negative for E gene amplification; (**b**) amplification curves of a sample positive for E gene amplification (orange line).

**Figure 2 diagnostics-11-02286-f002:**
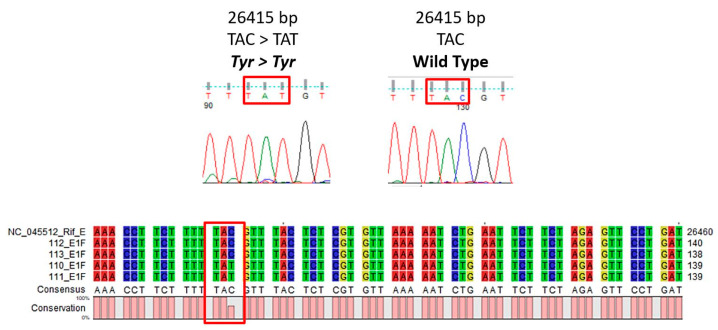
Chromatograms and sequences of some of the sequenced samples. Patients 110 and 111 have the synonymous mutation in the E gene, whereas the two control patients 112 and 113 have not.

**Figure 3 diagnostics-11-02286-f003:**
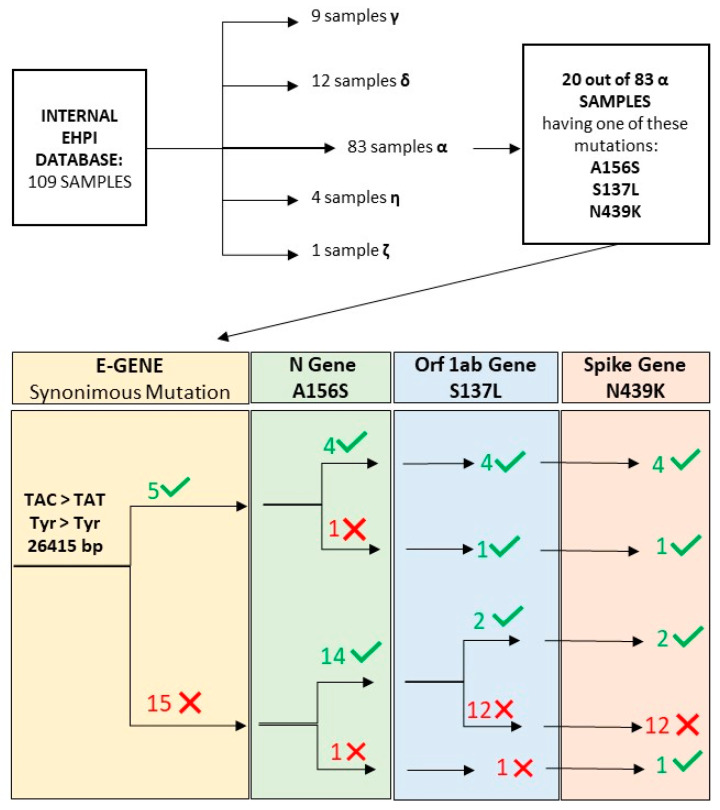
Dendrogram representing the 109 samples used for the analysis, the assigned lineages, and the characteristics of the 20 Alpha samples showing the three “extra-lineage” mutations identified in our work.

**Table 1 diagnostics-11-02286-t001:** Study sample NGS Average Coverage, coverage of four cluster mutations and Q-Score of NGS runs.

Patient	Average Coverage	c.26415 Coverage	A156S Coverage	N439K Coverage	S139L Coverage	Quality Score (Probability of Incorrect Base Call)
1	5505	2930	14,584	7345	4105	95% > Q_30_ (1 in 1.000)
2	5390	3025	13,304	8207	4625	
3	16,937	3125	75,924	2366	24,526	
4	13,607	1301	51,036	2505	26,154	
5	4732	194	-	7160	826	

**Table 2 diagnostics-11-02286-t002:** Primers used for E and Spike genes amplification in the PCR.

Gene		Primer Sequences 5′-3′	Position in Reference Genome	Amplicon Size (bp)
E gene				329
E1	F	TCTTTTTCTTGCTTTCGTGGT	26,298	
	R	ACAACCGTATCCGTTTAACA	26,627	
S gene				
FS-4	F	CCGCATCATTTTCCACTTTT	22,674	825
	R	CACGTCCGACAAATTATCCCC	23,499	
FS-5	F	TTTGGTTAAAAACAAATGTGTCAA	23,158	828
	R	AGGTAGTTTTGGTTCGTTCT	23,986	

**Table 3 diagnostics-11-02286-t003:** Function and frequency of the three mutations.

Mutation	Gene	GISAID Function	GISAID Frequency	EHPI Frequency	World Frequency
S137L	Orf 1ab	Not reported	0.12%	6.42%	0.21%
N439K	Spike	Surface Receptor Binding Antibody Recognition Sites—Viral Oligomerization Interfaces-Antigenic Drift	1.43%	7.33%	1.07%
A156S	Nucleocapsid	Viral Oligomerization Interface—Ligand Binding—Antigenic Drift	0.25%	21.20%	0.21%

**Table 4 diagnostics-11-02286-t004:** Correlation p-values calculated for each mutation (S137L, N439K and A156S). Above: comparison between the EHPI, Italy and World database populations. Below: comparison between the samples from our internal database with the mutated E gene and samples from our internal database with wild-type E gene. Statistically significant *p*-values: ≤0.05 *; ≤0.01 **; ≤0.001 ***; ≤0.0001 ****.

	EHPI vs. Italy	EHPI vs. World	Italy vs. World
S137L	0.290604089	1 × 10^7^—126 ****	2.38 × 10—11 ****
N439K	0.000146076 ***	8.99 × 10—10 ****	2.00 × 10—81 ****
A156S	3.95 × 10—53 ****	0 ****	0 ****
	E gene c.26415 C > T vs. Wild Type		
S137L	0.000433736 ***		
N439K	0.001565402 **		
A156S	0.389423696		

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
