# Peer review of "Low Represented Mutation Clustering in SARS-CoV-2 B.1.1.7 Sublineage Group with Synonymous Mutations in the E Gene"

_diagnostics, 2021, doi:10.3390/diagnostics11122286_

Round 1
Reviewer 1 Report
The manuscript identified a dropout of the E gene due to synonymous mutations and analyzed the frequencies of the mutations in different populations. This study is well organized and straightforward, and the data was presented clearly. These data are particularly important during this pandemic period. I would like to suggest the authors to add “SARS-CoV-2” in the title to make the paper more attractive to readers.
Author Response
The manuscript identified a dropout of the E gene due to synonymous mutations and analyzed the frequencies of the mutations in different populations. This study is well organized and straightforward, and the data was presented clearly. These data are particularly important during this pandemic period. I would like to suggest the authors to add “SARS-CoV-2” in the title to make the paper more attractive to readers.
We would like to thank the reviewer for the comment, we added “SARS-CoV-2” in the tittle as suggested: "Low represented mutation clustering in SARS-CoV-2 B.1.1.7 sublineage group with synonymous mutations in the E gene"
Reviewer 2 Report
Title: Low represented mutation clustering in the B.1.1.7 sublineage group with synonymous mutations in the E gene
Comments to the authors:
Summary: The paper talks about mutation cluster hypothesis and the work underlines the importance of territorial epidemiological surveillance by means of NGS.
The authors conducted sequencing analysis on six SARS-CoV-2-positive samples received from a Sicilian private analysis laboratory to test the mutation cluster hypothesis. The findings revealed the presence of a synonymous mutation (c.26415 C>T, TAC>TAT) in the E gene of all four samples showing the dropout in RT-qPCR. Also, authors state that these samples harbored three other mutations (S137L – Orf1ab; N439K – S gene; A156S – N gene) which had a very low diffusion rate worldwide suggesting that these mutations may be linked to each other and more common in a specific area than in the rest of the world. I recommend the publication of this article after consideration of a few minor comments below.
- Can the authors please check for punctuation errors in the manuscript.
- Can the authors elaborate the rational and significance of this study in a bit more detail.
- Can the authors describe about the E and S genes in a bit more detail in the introduction.
- The authors mentioned that the original primer pair does not produce amplification in RT-qPCR if the aforementioned synonymous mutation (TAC>TAT) was present as seen in the four samples tested. Can the authors clarify whether amplification is seen or not with the fifth sample from their internal database that also showed synonymous mutation.
- Can the authors justify their mutation cluster hypothesis given their analysis on just 4 SARS-CoV-2 positive samples.
- Can the authors comment on the evolutionary significance of these synonymous mutations given their low diffusion rate in the world.
Author Response
I recommend the publication of this article after consideration of a few minor comments below.
1. Can the authors please check for punctuation errors in the manuscript.
We would like thank the reviewer, we checked and corrected the manuscript.
2. Can the authors elaborate the rational and significance of this study in a bit more detail.
We would like thank the reviewer, we added a sentence that explains the rationale and significance of this study in closer details (lines 37-39 and 102-119).
3. Can the authors describe about the E and S genes in a bit more detail in the introduction.
We would like thank the reviewer, we added sentences about the E and S genes (lines 76-78 and 81-85).
4. The authors mentioned that the original primer pair does not produce amplification in RT-qPCR if the aforementioned synonymous mutation (TAC>TAT) was present as seen in the four samples tested. Can the authors clarify whether amplification is seen or not with the fifth sample from their internal database that also showed synonymous mutation.
The sample in question was not subjected to RT-qPCR. Given the result of the amplification of the four samples coming from the CDI (Centro Diagnostico Ionia) obtained with the mentioned kit, it is the lack of amplification of the E gene is to be associated precisely with the presence of the aforementioned mutations, therefore it would be redundant to test other samples with the same mutation in RT-qPCR. In support of this statement, it should be specified that the synonymous mutation c.26415, C> T, TAC> TAT falls precisely in the annealing region of the probe designed by this kit, which is why we did not find it necessary to further analyze this sample in RT-qPCR, in addition to Sanger and NGS.
5. Can the authors justify their mutation cluster hypothesis given their analysis on just 4 SARS-CoV-2 positive samples.
Our hypothesis of a mutational cluster comes from the fact that all four samples harbored a very similar set of mutations besides the four "main" mutations, and they all came from a very specific geographic location. Our hypothesis gains plausibility when the observed frequency of these mutations in the rest of the country is low. Unfortunately, as already mentioned, we would have liked to better follow the evolution of the cluster, but the summer holidays and the arrival of a large number of tourists in Sicily did not allow us to proceed as we meant to.
6.Can the authors comment on the evolutionary significance of these synonymous mutations given their low diffusion rate in the world.
A large body of studies aiming to identify and understand non-synonymous mutations of the SARS-CoV-2 genome is reported in the literature; however, very few analyses on synonymous mutations are reported. A peculiarity of these mutations is that they are not subject to selective pressure, unlike non-synonymous mutations (PMID: 8849908). For this reason, they can be useful in calculating the divergence time between different lineages. Although these mutations do not alter amino acid sequences, some authors suggest that they can still modify the protein structure. The evolutionary importance of these mutations has been demonstrated in recent years; in fact, at a molecular level, synonymous mutations can alter the pre-mRNA splicing (PMID: 24630730), the availability of t-RNA (PMID: 15448185), the translational speed (PMID: 32888056) and many other factors almost always involved in protein synthesis (PMCID: PMC8132620).
Reviewer 3 Report
The study by Paolo Giuseppe Bonacci et al. about the "Low represented mutation clustering in the B.1.1.7 sublineage group with synonymous mutations in the E gene" is an interesting one. It highlights the possibility of territorial epidemiological surveillance by means of NGS and the sequencing of samples with clinical and or technical particularities, e.g., post-vaccine infections or RT-qPCR amplification failures, to allows for the early identification of these SNPs.
But, there are certain required experimental strengthening to be done before taking it forward. This includes:
- One particular RT-PCT kit was used for SARS-CoV-2 RT-PCR and the dropout of the E gene expression was observed. There has been publications highlighting the role of SARS-CoV-2 genomic mutations/SNPs within the probe binding region which may role in sub-optimal RT-PCR performance. The E gene dropout hypothesis needs to be strengthened with atleast couple of other RT-PCR reagents from different company before reaching the above conclusion.
- Regarding the samples number, it is quite small (6 samples) for inference like this. Additional samples from the same geographical area and matching time of sampling should be done to strengthen this observation.
- The comparison within the EHPI database showed one sample with matching synonymous mutation (c.26415 C> T, TAC> TAT) in the E gene. What is not clear from the presented results is that i) whether it also carried the other 3 mutations, ii) did this samples also showed drop for the E gene during RT-PCR using the same kit?, iii) what was the comparative performance of the samples with other RT-PCR kits?, iv) was this mutation or combination of mutations present in any other geographical location/s of Italy?
- What is the geographical location of the samples in GISAID with matching mutation/s? I understand the low frequency aspect but whether it is concentrated in and around Italy or distributed globally?
- The sequencing details and associated stats need to be mentioned especially around the mutations we are talking about. To specify - NGS data.
The addressing of the above suggestions would help take forward the hypothesis explored in the manuscript.
Best wishes,
Author Response
But, there are certain required experimental strengthening to be done before taking it forward. This includes:
1. One particular RT-PCT kit was used for SARS-CoV-2 RT-PCR and the dropout of the E gene expression was observed. There has been publications highlighting the role of SARS-CoV-2 genomic mutations/SNPs within the probe binding region which may role in sub-optimal RT-PCR performance. The E gene dropout hypothesis needs to be strengthened with atleast couple of other RT-PCR reagents from different company before reaching the above conclusion.
The ”Centro Diagnostica Ionia” mentioned in the paper had previously analyzed the four samples with the E-dropout with another kit (SARS-CoV-2 Assay Allplex Seegene, Ref: RV10428X), different from the one reported in the article. In this case, the amplification of the E gene gave a positive result, supporting that the synonymous mutation c.26415, C> T, TAC> TAT in the aforementioned gene only affects the gene amplification if processed with the Adaltis kit, which however correctly identified the samples positivity for SARS-CoV-2. It should be specified that the guidelines for SARS-CoV-2 epidemiological surveillance established in Sicily provide that samples showing fluorescence artifacts during RT-qPCR be sequenced by Next Generation Sequencing, which is why the aforementioned diagnostic center has contacted our research group.
2. Regarding the samples number, it is quite small (6 samples) for inference like this. Additional samples from the same geographical area and matching time of sampling should be done to strengthen this observation.
Thanks for the suggestion, we agree that the number of samples is not large. We would like to specify that only these four samples have the mutational cluster we were interested in, while the others have not. Furthermore, in this case the number of samples does not depend on the experimental design, but it is the genetic result that determines the number. The total amount of samples analyzed is actually 109 (the Internal Database), all of which were compared with the entire GISAID database. Despite this, we continued our epidemiological surveillance for the identification of other samples with the E-dropout in collaboration with the Centro Diagnostico Ionia. Unfortunately, it was epidemiologically difficult to monitor this cluster, because the cluster identification coincided with the beginning of the summer holidays and with the end of the lockdown in our country, so the cluster was probably supplanted by the interference of a series of lineages from all over the world, Sicily being a very popular tourist destination.
3. The comparison within the EHPI database showed one sample with matching synonymous mutation (c.26415 C> T, TAC> TAT) in the E gene. What is not clear from the presented results is that i) whether it also carried the other 3 mutations, ii) did this samples also showed drop for the E gene during RT-PCR using the same kit?, iii) what was the comparative performance of the samples with other RT-PCR kits?, iv) was this mutation or combination of mutations present in any other geographical location/s of Italy?
i. The sample already included in our internal database, beside the synonymous mutation c.26415, C> T, TAC> TAT in the E gene, also presented the mutations N439K (Spike gene) and S137L (Orf 1ab gene), but not the A156S mutation (N gene).
ii. The sample in question was not subjected to RT-qPCR. Given the result of the amplification of the four samples coming from the CDI obtained with the mentioned kit, it is the lack of amplification of the E gene is to be associated precisely with the presence of the aforementioned mutations, therefore it would be redundant to test other samples with the same in RT-qPCR. In support of this statement, it should be specified that the synonymous mutation c.26415, C> T, TAC> TAT falls precisely in the annealing region of the probe designed by this kit, which is why we did not find it necessary to further analyze this sample in RT-qPCR, in addition to with Sanger and NGS.
iii. The ”Centro Diagnostica Ionia” mentioned in the paper had previously analyzed the four samples with the E-dropout with another kit (SARS-CoV-2 Assay Allplex Seegene, Ref: RV10428X), different from the one reported in the article. In this case, the amplification of the E gene gave a positive result, supporting that the synonymous mutation c.26415, C> T, TAC> TAT in the aforementioned gene only affects the gene amplification if processed with the Adaltis kit, which however correctly identified the samples positivity for SARS-CoV-2. It should be specified that the guidelines for SARS-CoV-2 epidemiological surveillance established in Sicily provide that samples showing fluorescence artifacts during RT-qPCR be sequenced by Next Generation Sequencing, which is why the aforementioned diagnostic center has contacted our research group.
iv. All mutations, if taken individually, can be monitored on the GISAID database, which confirmed their presence both in Italy and in the rest of the world, although with a significantly lower frequency than in our sample. However, no information of this kind can be given as regards the clusters, as GISAID does not search for combinations of mutations, and it is therefore impossible to make a relationship between the clusters.
4. What is the geographical location of the samples in GISAID with matching mutation/s? I understand the low frequency aspect but whether it is concentrated in and around Italy or distributed globally?
The single mutations are reported from sequences all over the world in a fairly homogeneous way, while the combination of the mutations reported by us seems to be a local phenomenon, localized in Southern Italy and especially in eastern Sicily. For this reason, we also considered it necessary to carry out a statistical study, from which it emerges that the cluster of mutations we identified is "closer" to Italy than to the rest of the world.
5. The sequencing details and associated stats need to be mentioned especially around the mutations we are talking about. To specify - NGS data.
|
Patient |
Average Coverage |
c.26415 Coverage |
A156S Coverage |
N439K Coverage |
S139L Coverage |
Quality Score (Probability of Incorrect Base Call) |
|
1 |
5505 |
2930 |
14584 |
7345 |
4105 |
95% > Q30 (1 in 1.000) |
|
2 |
5390 |
3025 |
13304 |
8207 |
4625 |
|
|
3 |
16937 |
3125 |
75924 |
2366 |
24526 |
|
|
4 |
13607 |
1301 |
51036 |
2505 |
26154 |
|
|
5 |
4732 |
194 |
- |
7160 |
826 |
This table with different NGS quality parameters has been inserted in the text. If you think this might be appropriate, we can supply the FASTQ files deriving from the instrument run for the corresponding samples.
Reviewer 4 Report
This manuscript is well written, although the authors could improve on the introduction to reflect the aim and objective of the study. and also address the gap in knowledge in genomic surveillance The discussion also can be improved to reflect the outcome of the results. They can also add a concluding statement after the discussion to say how the study results contribute new knowledge to genomic surveillance.
Author Response
This manuscript is well written, although the authors could improve on the introduction to reflect the aim and objective of the study. and also address the gap in knowledge in genomic surveillance The discussion also can be improved to reflect the outcome of the results. They can also add a concluding statement after the discussion to say how the study results contribute new knowledge to genomic surveillance.
We would like thank the reviewer, we expanded on these aspects as suggested, lines 274-300 and 333-343.
Round 2
Reviewer 3 Report
Revised manuscript can be accepted.